# Uncovering changes in microbiome profiles across commercial and backyard poultry farming systems

Muhammed Shafeekh Muyyarikkandy,[1] Jessica Parzygnat,[1] Siddhartha Thakur[1]

**ABSTRACT** The microbiome profiles of poultry production systems significantly impact bird health, welfare, and the environment. This study investigated the influence of broiler-rearing systems on the microbiome composition of commercial and backyard chicken farms and their environment over time. Understanding these effects is vital for optimizing animal growth, enhancing welfare, and addressing human and environmental health implications. We collected and analyzed various samples from commercial and backyard farms, revealing significant differences in microbial diversity measurements between the two systems. Backyard farms exhibited higher alpha diversity measurements in soil and water samples, while commercial farms showed higher values for litter and feeder samples. The differences in microbial diversity were also reflected in the relative abundance of various microbial taxa. In backyard farms, Proteobacteria levels increased over time, while Firmicutes levels decreased. Campilobacterota, including the major poultry foodborne pathogen *Campylobacter*, increased over time in commercial farm environments. Furthermore, *Bacteroides*, associated with improved growth performance in chickens, were more abundant in backyard farms. Conversely, pathogenic *Acinetobacter* was significantly higher in backyard chicken fecal and feeder swab samples. The presence of *Brevibacterium* and *Brachybacterium*, associated with low-performing broiler flocks, was significantly higher in commercial farm samples. The observed differences in microbial composition and diversity suggest that farm management practices and environmental conditions significantly affect poultry health and welfare and have potential implications for human and environmental health. Understanding these relationships can inform targeted interventions to optimize poultry production, improve animal welfare, and mitigate foodborne pathogens and antimicrobial resistance risks.

**IMPORTANCE** The microbiome of poultry production systems has garnered significant attention due to its implications on bird health, welfare, and overall performance. The present study investigates the impact of different broiler-rearing systems, namely, commercial (conventional) and backyard (non-conventional), on the microbiome profiles of chickens and their environment over time. Understanding the influence of these systems on microbiome composition is a critical aspect of the One-Health concept, which emphasizes the interconnectedness of animal, human, and environmental health. Our findings demonstrate that the type of broiler production system significantly affects both the birds and their environment, with distinct microbial communities associated with each system. This study reveals the presence of specific microbial taxa that differ in abundance between commercial and backyard poultry farms, providing valuable insights into the management practices that may alter the microbiome in these settings. Furthermore, the dynamic changes in microbial composition over time observed in our study highlight the complex interplay between the poultry gut microbiome, environmental factors, and production systems. By identifying the key microbial players and their fluctuations in commercial and backyard broiler production systems, this research

Address correspondence to Muhammed Shafeekh Muyyarikkandy, mmuyyar@ncsu.edu.

The authors declare no conflict of interest.

See the funding table on p. 12.

offers a foundation for developing targeted strategies to optimize bird health and welfare while minimizing the potential risks to human and environmental health. The results contribute to a growing body of knowledge in the field of poultry microbiome research and have the potential to guide future improvements in poultry production practices that promote a sustainable and healthy balance between the birds, their environment, and the microbial communities they host.

**KEYWORDS** poultry microbiome, broiler farming, relative abundance, marker taxa, temporal changes

Poultry meat is one of the most consumed meats around the world. However, it is also a major reservoir for major foodborne pathogens. The gastrointestinal tracts of chickens are relatively short and possess a quicker transit time than mammals (1). The gastrointestinal tract contains diverse microbiota, and their metabolites significantly enhance nutrient absorption and immune function (2). Closely related hosts seem to share a more similar microbiome composition than distantly related hosts (3, 4). However, environmental factors contribute to significant alterations in the host microbiome, and the rearing system may affect the chicken gastrointestinal microbiome (5). Stress due to change in rearing conditions, such as heat stress, lead to gut dysbiosis and reduce alpha diversity of microbiome composition (6). Globally, antibiotics have been used for growth promotion, treatment, prevention, and control of diseases, and antibiotic use could result in gut dysbiosis in chickens (7). It is important to understand factors that affect a balanced gut microbiome, as it is essential for maintaining a healthy gut environment for normal functioning of chicken metabolism.

Consumers in the United States are shifting their preference for non-conventionally raised meat and eggs (8). Traditionally, poultry has been reared in an intensive indoor floor system. Pasture-raised or free-range chicken production continues to grow in popularity and is linked economically to local markets (9). Moreover, organically raised antibiotic-free meat and eggs are gaining more popularity globally (10). However, food safety, nutrition, meat quality, and processing are major concerns in pasture-raised poultry (11). Further research is needed to understand environmental issues better and the food safety of pasture-raised poultry as local retail markets have become more prominent (12). Moreover, there is an urgent need to optimize the health and performance of pasture-raised chickens (13). Additionally, there are challenges and benefits associated with pasture flock broiler production. For example, microbial communities in birds reared in nonconventional farms harbor microorganisms with potential probiotic properties. However, environmental stressors such as heat stress and parasites can contribute to bird health (13).

Poultry production can have a significant impact on the environmental microbiome. The introduction of large numbers of poultry into an area can increase nutrient levels and alter the existing microbial communities in the environment (14). Poultry waste is a major source of nutrients such as nitrogen and phosphorus, which can lead to eutrophication (an overabundance of plant growth) in aquatic ecosystems (15). This increased nutrient availability provides ideal conditions for certain bacteria species to flourish while reducing populations of other microbes that are not able to compete under these altered conditions (16). Additionally, changes in temperature and water pH caused by poultry operations may further affect microbial diversity within affected environments (17).

Most research addressing poultry production's impact on environmental impacts has focused on conventional production systems (12). Minimal studies investigate pasture-raised poultry production's impact on environmental impact. This study investigates the effects of conventional and pasture-raised broiler production systems on poultry and environment microbiomes. To achieve this, we sampled five commercial (conventional) and five pasture-raised (non-conventional) broiler farms three times throughout the production period. Fecal and environmental samples were collected, and their microbiome profiles were analyzed to delineate the compositional differences.

## MATERIALS AND METHODS

### Farm demographics and operational practices

The study involved sampling from five commercial farms and five backyard farms, adhering to the minimum inclusion criteria of raising at least 100 chickens without the usage of antibiotics as growth promoters. The anonymous data were safeguarded by a coding system, assuring confidentiality of farm information. Our sampling protocol varied according to the type of farm; backyard farms were visited thrice during the production cycle, specifically on days 10, 31, and 52, while commercial farms were visited on days 10, 24, and 38, reflecting the more rapid pace of broiler production in these environments. The chosen time points likely represent key stages in the broiler's growth and health, allowing the study to capture meaningful changes in the microbiome across their accelerated growth period. The commercial farms, employing an intensive production system, reared thousands of birds indoors, with new litter added prior to each stocking cycle. Composting was not implemented in these settings. Conversely, backyard farms, which usually raise between 100 and 1,000 birds at a time, practiced on-site composting and a daily rotation of pens on the pasture. Furthermore, multiple livestock species were maintained in the backyard farms. A marked contrast was observed in biosecurity measures between the two farm types. Commercial farms implemented rigorous procedures such as tire washes, foot dips, and personal protective equipment usage. However, biosecurity practices in backyard farms were more varied, with gum boots being the most common protective gear used. Lastly, the typical production duration in commercial farms was 5–6 weeks, with backyard farms taking a slightly longer time of 8–10 weeks. This may reflect the differences in breed genetics and production strategies between the two settings. However, data collection did not include a baseline microbiome sample at day 0 (chick placement), an aspect that could have provided valuable insight into initial conditions. The specific rationale for the chosen time points of farm visits pertained to the different production schedules of commercial and backyard farms.

### Sample collection

This study aimed to investigate the effect of the broiler production system on the microbiome. Various samples were collected from five backyard and five commercial farms in an effort to assess both bird and environmental microbiomes. The samples collected from each farm during each visit include ten fecal samples, five soil samples, five litter and/or compost samples, and six swabs split between waterers and feeders. The collected samples were snap frozen until processing. Each backyard farm was sampled at three time points: 10, 31, and 52 d of production. Since the commercial farms rear chickens for a shorter period than backyard farms, sampling times were 10, 24, and 38 d of production.

### DNA extraction and sequencing

Total DNA was extracted using DNeasy PowerLyzer Powersoil Kit (Qiagen, MD, United States), and all DNA samples were kept at −20°C until further analysis using the Illumina MiSeq sequencing platform. Each sample's total DNA was fragmented and tagged with sequencing adapters using Nextera XT DNA Library Preparation Kit (Illumina, San Diego, CA). As previously described, the gene-specific sequence targeted the 16S V3 and V4 regions (18). Metagenomic sequencing yielded an average of ~1 million reads/sample. Following sequencing, all reads were quality assessed, and the paired-end was merged. The PCR products were pooled in equal proportions based on their molecular weight and concentrations. Illumina DNA library was prepared using the calibrated Ampure beads-purified PCR products with a final loading concentration of 3 pM and 25% PhiX control. Sequencing was performed using MiSeq V3 Kit on a MiSeq at 300 bp, 60 cycles, and in paired-end mode.

## Statistical and network analysis

A customized workflow of the Divisive Amplicon Denoising Algorithm (DADA2 v1.10) package was used to process the sequencing data (19). A table of amplicon sequence variants (ASVs) was obtained by denoising using a customized workflow. The workflow includes quality control, primer removal, denoising, and taxonomy assignments. Cutadapt was used to trim forward and reverse primers, and reads were merged to create a paired-end sequence. Quality checking (QC) was performed to check the read lengths and the quality of the joined sequences. The final output was in the form of a BIOM table with all the information regarding the sequences, abundances of ASVs, and the assigned taxa information.

All analyses were performed using the MicrobiomeAnalyst, platform. MicrobiomeAnalyst is a comprehensive statistical, visual, and meta-analysis tool for microbiome data that utilizes the MicrobiomeAnalyst R package for statistical analysis and graphical outputs. Low abundant features were filtered based on a 20% prevalence filter and a minimum count set at 4. The low variance features were removed based on the interquartile range with a cutoff range of 10%. All samples were rarified to even sequencing depth based on the sample having the lowest sequence depth, and the remaining features were normalized using the total sum scaling (TSS) method. Alpha diversity was calculated using the observed species, Chao1, Simpson, ACE, Fisher and Shannon indexes with a $t$-test or ANOVA statistical analysis. For the beta diversity and significant testing, the PCoA ordination method was used with the Bray-Curtis index at the species level, and PERMANOVA was applied as the statistical method. The core microbiome analysis was conducted at the genus and phylum levels with a relative abundance of 0.01 and a sample prevalence of 20.

Moreover, a pattern search was conducted to identify microbiome patterns at all growth stages using the SparCC correlation at the order level. Linear discriminant analysis (LDA) was used for the biomarker analysis. The platform performs a non-parametric factorial Kruskal-Wallis sum-rank test to identify features with significant differential abundance considering the class of interest, followed by LDA to calculate the effect size of each differentially abundant feature. The heat tree was plotted using a Wilcoxon p-value cutoff −0.05. The features are considered significant depending on their adjusted p-value. The default Wilcoxon p-value cutoff is −0.1, and the LDA score is 2.0.

## RESULTS

### Alpha diversity measurements

Different alpha diversity measurements were calculated between the backyard and commercial farms (Table 1). The alpha diversity measurements of fecal samples were very similar between commercial and backyard farms, and no significant difference was detected for any alpha diversity indexes. The observed species defines the species richness by measuring the number of different species per sample. The observed species index was significantly higher in commercial farms for the soil and water samples, and it was higher in litter samples of backyard farms. However, there was no significant difference in feeder samples. The Chao1 index gives more weight to rare species by considering the ratio of singletons to doubletons in addition to species richness. The Chao1 and ACE measurements were significantly higher in soil and water samples of backyard samples; however, they were higher in litter and feeder samples of commercial farms. Regarding diversity, not only the qualitative number of species but also the actual abundance of observed species must be considered. The Shannon index takes into account both species richness and evenness. The Simpson index measures diversity by considering the relative abundance of each species and the number of species present. The Simpson index of litter and water samples of backyard farms was significantly higher than those in commercial farms. Fisher's alpha diversity index was higher in backyard farm soil, feeder, and water samples. Overall, alpha diversity measurements of soil and

**TABLE 1**  Alpha diversity measurements of various samples collected from commercial and backyard broiler farms

| Alpha diversity | Fecal | | Soil | | Litter | | Feeder | | Waterer | |
|---|---|---|---|---|---|---|---|---|---|---|
| | Backyard | Commercial | Backyard | Commercial | Backyard | Commercial | Backyard | Commercial | Backyard | Commercial |
| Observed | 396 | 404 | 1410$^a$ | 992 | 256 | 267$^a$ | 417 | 533 | 264$^a$ | 194 |
| Shannon | 3.77 | 3.67 | 5.58 | 6.02$^a$ | 4.27$^a$ | 3.83 | 3.91 | 4.11 | 3.70$^a$ | 2.92 |
| Simpson | 0.93 | 0.93 | 0.97 | 0.98 | 0.96$^a$ | 0.94 | 0.92 | 0.93 | 0.91$^a$ | 0.78 |
| Chao1 | 451 ± 14 | 463 ± 19 | 1,656 ± 34$^a$ | 1,179 ± 30 | 285 ± 11 | 375 ± 29$^a$ | 457 ± 13 | 594 ± 16$^a$ | 342 ± 23$^a$ | 232 ± 14 |
| ACE | 451 ± 10 | 453 ± 10 | 1,656 ± 18$^a$ | 1,176 ± 15 | 285 ± 8 | 383 ± 10$^a$ | 453 ± 10 | 589 ± 11$^a$ | 341 ± 9 | 233 ± 7$^a$ |

$^a$indicates that the value is significantly higher in that production system.

water samples were higher in backyard farms, and values of litter and feeder samples were higher in commercial farms.

## Beta diversity and relative abundance

The relative abundance of top microbes with higher proportions is shown in Fig. 1A. The fecal samples from backyard farms were found to have more Firmicutes, Bacteriodota, Desulfobacteria, Synergitota, Fusobacteriota, and Campilobacterota. However, there were more Actinobacteriodota in commercial farms compared to backyard farms. Soil samples from both backyard and commercial farms exhibited more richness and evenness, and Proteobacteriota was the major phylum in both groups. The proportion of Firmicutes and Bacteriodota was higher in backyard farm soil, while Actinobacteriota and Aidobacteriota were higher in commercial farms. Litter samples of commercial farms were less diverse, and Firmicutes and Actinobateriodota account for most of the microbial composition. However, the litter samples of backyard farms constituted significantly higher proportions of Proteobacteria and Bacteroidota. Feeder and waterer samples of both commercial and backyard farms contain higher proportions of Firmicutes. Proteobacteria was the second highest proportionate phylum in feeders of backyard farms, while it was Actinobacteriota in commercial farms. The proportion of Proteobacteriota and Bacteroidota was significantly higher in backyard farms compared to commercial farms. Feature-level dissimilarities between the microbial composition of commercial and backyard farms were tested using the PCoA ordination method using the Bray-Curtis distance index and PERMANOVA as a statistical method. PCoA results are displayed in Fig. 1B. Litter and feeder samples were clearly separated, indicating prominent differences in their microbiome composition of commercial and backyard farms. Fecal samples of commercial farms (blue dots) were shifted to the left, indicating a compositional difference. However, the soil samples of backyard farms (red dots) were clustered within the commercial farms. Waterer samples of backyard farms formed a compact cluster inside commercial farms; however, significant overlap was detected.

## Changes in microbiome composition depending on poultry farming systems

Significant taxa are ranked in decreasing order by their Linear Discriminant Analysis Effect Size (LEfSe) scores (Fig. 2). The LEfSe algorithm is used to discover and interpret metagenomics data by employing the Kruskal-Wallis rank sum test. LEfSe detects features with significant differential abundance, followed by LDA to evaluate the effect size of differentially abundant features. Differentially abundant features were detected in different samples collected from commercial and backyard farms. Out of the top 15 differentially abundant genera in fecal samples, eight genera, *Bacteroides, Megamonas, Desulfovibrio, Acinetobacter, Phasobacterium, Tepidibacter, Synerggistes,* and *Fusobacterium* were significantly higher in backyard farms, while *Nocardioposis, Jeotgalicoccusl, Atopostipes, Salinicoccus, Brevibacterium, Brachybacterium,* and *Staphylococcus* were significantly higher in commercial farms.

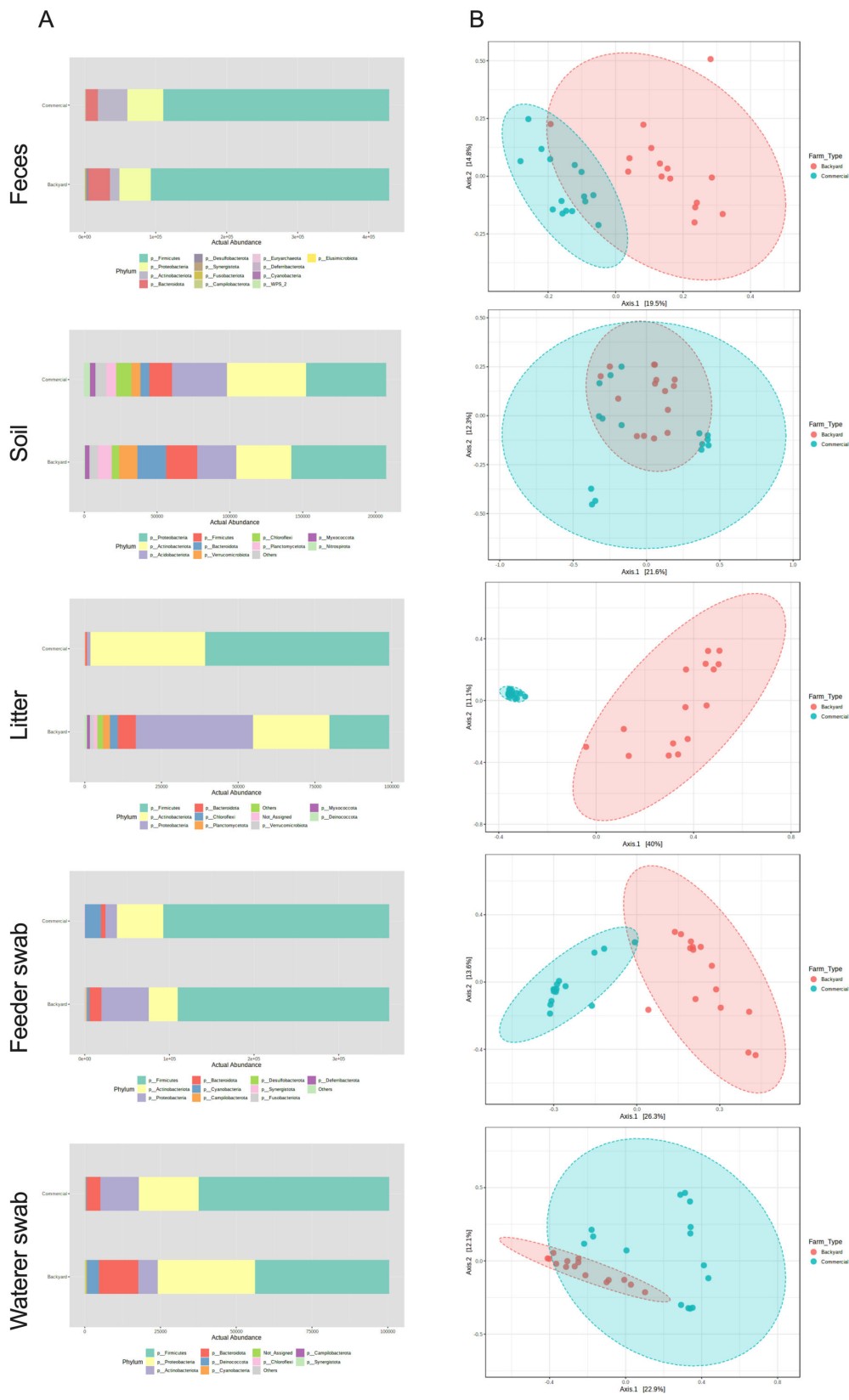

**FIG 1** Microbiome composition and beta diversity measurements of different samples from commercial and backyard broiler production systems. (A) The panel on the left shows relative abundance of major phyla, and (B) the panel on the right shows beta diversity measurements.

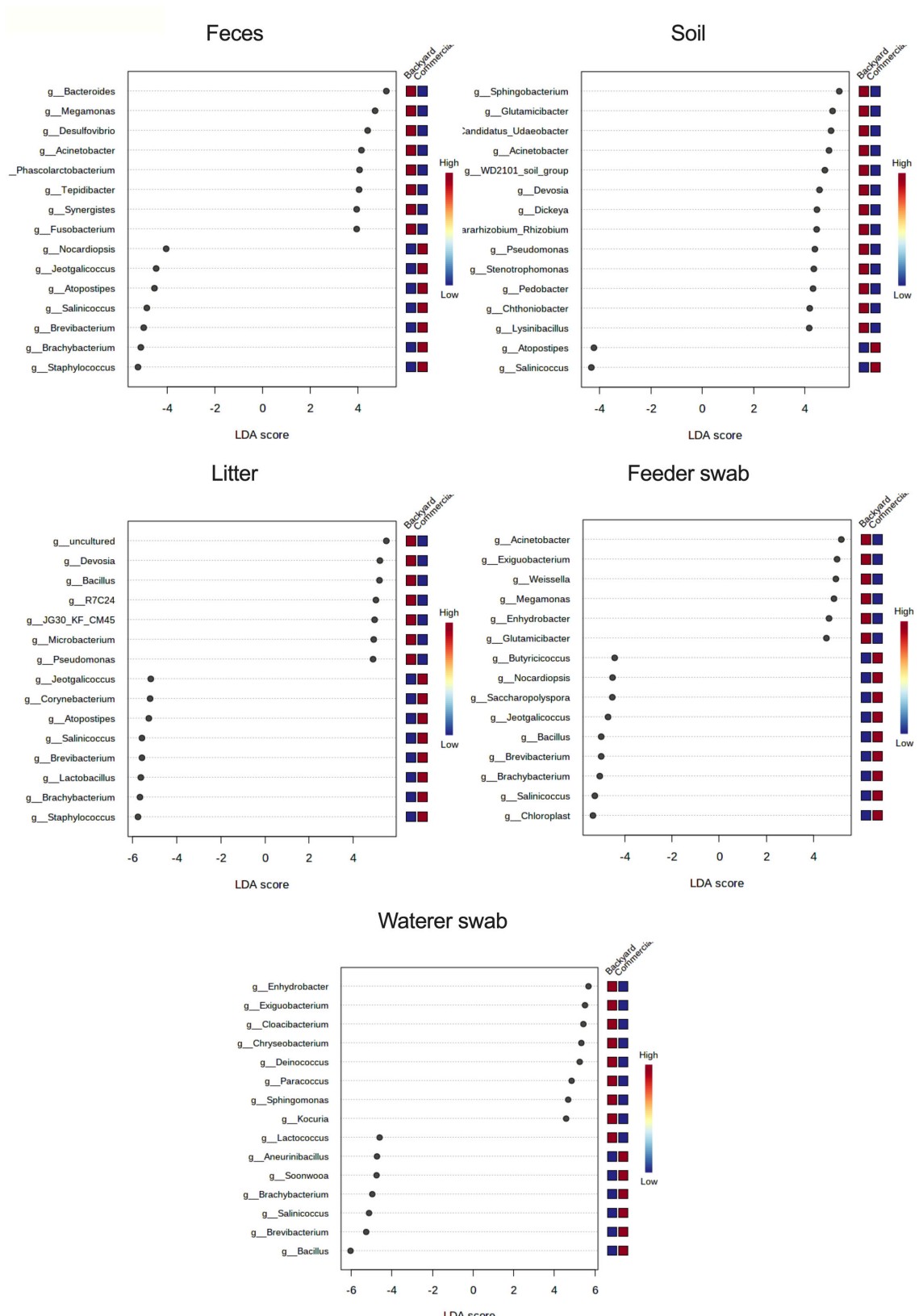

**FIG 2** Linear Discriminant Analysis based on Effect Size scores. The placement of feature toward the right indicates that it is higher in backyard farms, and the placement of feature toward the left indicates that it is higher in commercial farms.

## Microbiome clustering and differences in the abundance of community composition

A hierarchical clustering analysis was conducted for all the samples from commercial and backyard farms. The dendrogram in Fig. 3A and B show the Ward clustering based on Bray-Curtis distances of all the samples based on sample types. Clustering showed different patterns in commercial and backyard farms. In commercial farms, sample-dependent clustering was observed. In backyard farms, soil and litter samples were clustered together. Taxonomic differences in microbial communities of commercial and backyard farms at the family level are represented in the heat tree (Fig. 3C). In Fig. 3C, blue and red indicate that corresponding taxa are lower and higher in commercial farms compared to backyard farms. It is noted that Pectobacteriaceae and Moraxellaceae are lower in commercial farms compared to backyard farms. However, *Bacillaceae*, *Clostridiaceae*, Carnobacteriaceae, Staphylococcaceae, Corynebacteriaceae, and Ruminococcaceae are higher in commercial farms. Overall, Cyanobacteria, Firmicutes, Desulfobacteria, and Actinobacteria are significantly higher in commercial farms, while Proteobacteria, Fusobacteria, Bacteriodota, Campilobacteriota, and Acidobacteriota are significantly higher in backyard farms (Fig. 3D).

## Changes in microbiome profiles over time

Hierarchical clustering and heatmap visualize differences in abundance at the genus level, which is clustered using the Ward algorithm and Euclidean distance measurement (Fig. 4A). In backyard farms, the levels of Firmicutes in soil, litter, and swab samples tend to decrease with time; they were lower during the third visit compared to the first visit. Moreover, the levels of Proteobacteria increased with time in the sample from backyard farms. In backyard farms, Proteobacteria, Euryarchaeota, Vernucomicrobiota, and Chloroflexi exhibited a temporal increase, with the lowest abundance during the first visit and the highest abundance during the third visit (Fig. 4B). However, Campilobacterota and Firmicutes levels were highest during the first visit and lowest during the third visit. However, Campilobacterota levels were lower during the first and second visits to commercial farms. In addition, Desulfobacterota abundance increased consistently over time in commercial farms (Fig. 4C).

## DISCUSSION

The microbiome of poultry production systems is an important area of research, as it can provide insights into the health and welfare of the birds being raised. This study aimed to investigate how the broiler-rearing system structures the microbiome profiles of commercial (conventional) and backyard (non-conventional) chicken and their environment over time. Given the importance of the microbiome for animal, human, and environmental health, identifying taxa that differ significantly in abundance according to broiler production system may be particularly important as a previously underappreciated aspect of One-Health. The population of Proteobacteria in backyard farms demonstrated a rising trend over time, whereas the number of Firmicutes showed a decreasing trajectory. In commercially raised poultry, there was an upward trend in the prevalence of Campilobacterota, which includes *Campylobacter*, a primary pathogen causing foodborne diseases originating from poultry. Interestingly, a greater abundance of *Bacteroides*, which is associated with enhanced growth rates in chickens, was observed in backyard farms. However, a drawback was the significant increase in the concentration of potentially harmful *Acinetobacter* in samples from both backyard chicken feces and feeder swabs. Commercial farms, on the other hand, saw a marked increase in *Brevibacterium* and *Brachybacterium*, bacteria associated with broiler flocks that perform poorly. The implications of these findings can significantly impact the poultry farming industry. The increasing levels of Proteobacteria and *Bacteroides* in backyard farms might suggest a potential for improved growth performance in these environments. However, the higher presence of *Acinetobacter* poses a health risk that needs to be managed. On

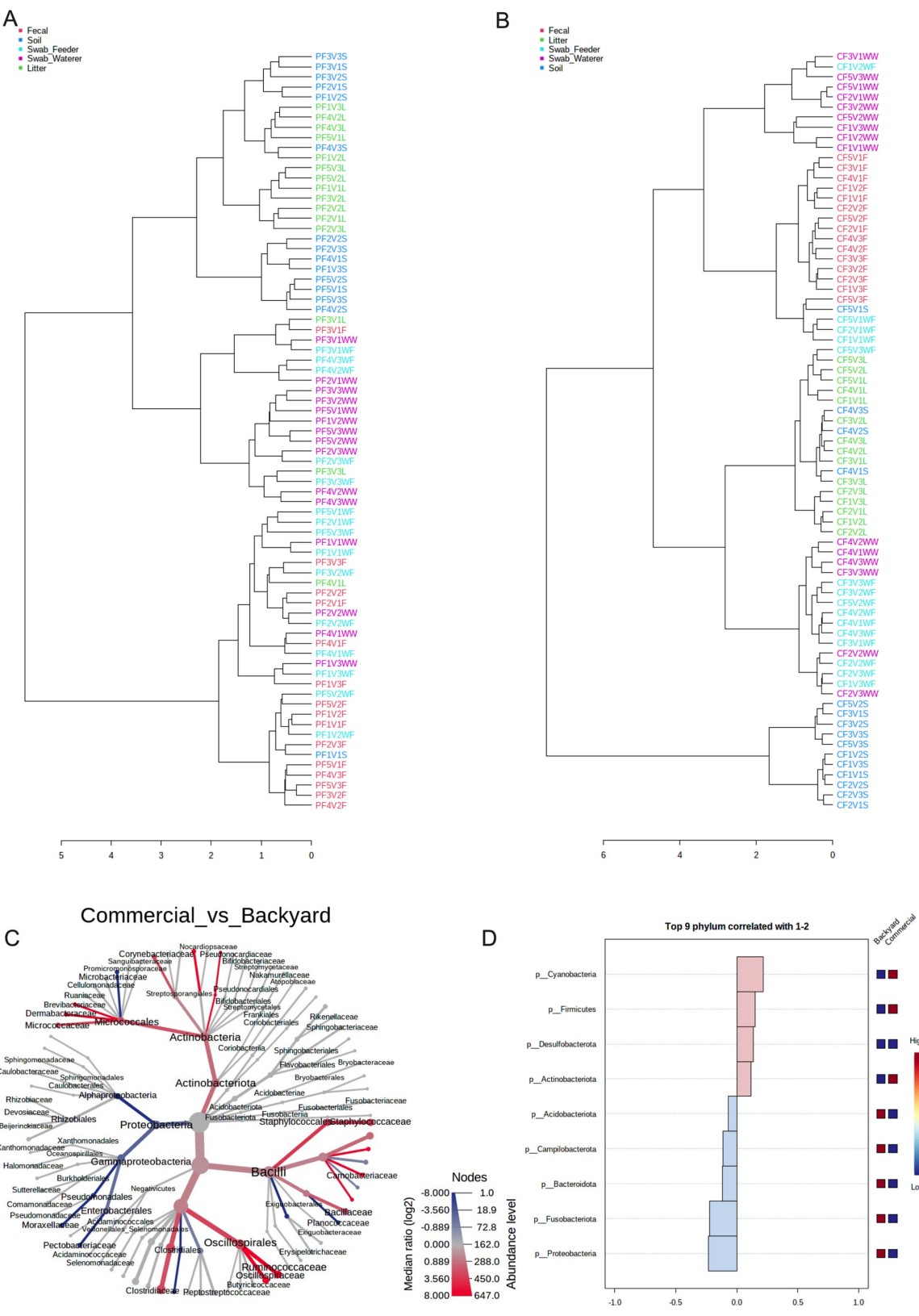

**FIG 3** Hierarchical clustering and relative abundance. (A) Dendrogram showing hierarchical clustering of commercial farm samples. (B) Dendrogram showing hierarchical clustering of backyard farm samples. (C) Heat tree representing the difference in taxonomic abundance. Blue and red indicate that corresponding taxa are lower and higher in commercial farms compared to backyard farms. (D) Correlation of top phyla associated with commercial and backyard farms.

the commercial side, the increasing presence of the foodborne pathogen *Campylobacter* requires vigilance and proactive strategies to prevent potential outbreaks. The presence of *Brevibacterium* and *Brachybacterium*, indicators of low-performing broiler flocks, might suggest a need for improved farming practices to enhance the productivity and overall health of the flocks in commercial farm environments.

The broiler production system can alter the taxonomic composition of the microbiome in commercial and backyard poultry farms by promoting certain microbial groups while suppressing others (20). In commercial settings, modern management practices such as proper sanitation, cleaning, disinfection protocols, and vaccination use have drastically reduced or eliminated many species of commensal bacteria traditionally associated with chickens, leading to a decrease in biodiversity and an increased prevalence of avian pathogenic organisms such as *Salmonella* (21). Simultaneously, some beneficial microbes that are involved in nutrient utilization or immune modulation have been found to be enriched through these measures (22). Backyard flocks usually keep a more diverse microbiota compared to intensively reared flocks due to fewer sanitary interventions; however, this diversity is still influenced by farm size and environmental conditions (20).

In this study, we observed a significant relationship between the type of broiler production system and the microbial composition of environmental and fecal samples. The type of poultry production system significantly affected the overall profile of the microbial community measured at phylum, family, genus, and feature levels. Diversity measurements such as species richness, evenness, and abundance were also significantly different according to farm type. However, we did not observe any significant difference in alpha diversity measurements of fecal samples from conventional and pasture-raised broilers. A study by Rama shows that the alpha diversity measurements of conventional and non-conventional farms did not differ until day 43; however, they started diverging after that with higher diversity measurements exhibited in non-conventional farms (23). In this study, the slaughter age of broilers raised in conventional farms was around 42–49 d, and our last sampling time was day 38 in conventional farms. However, the alpha

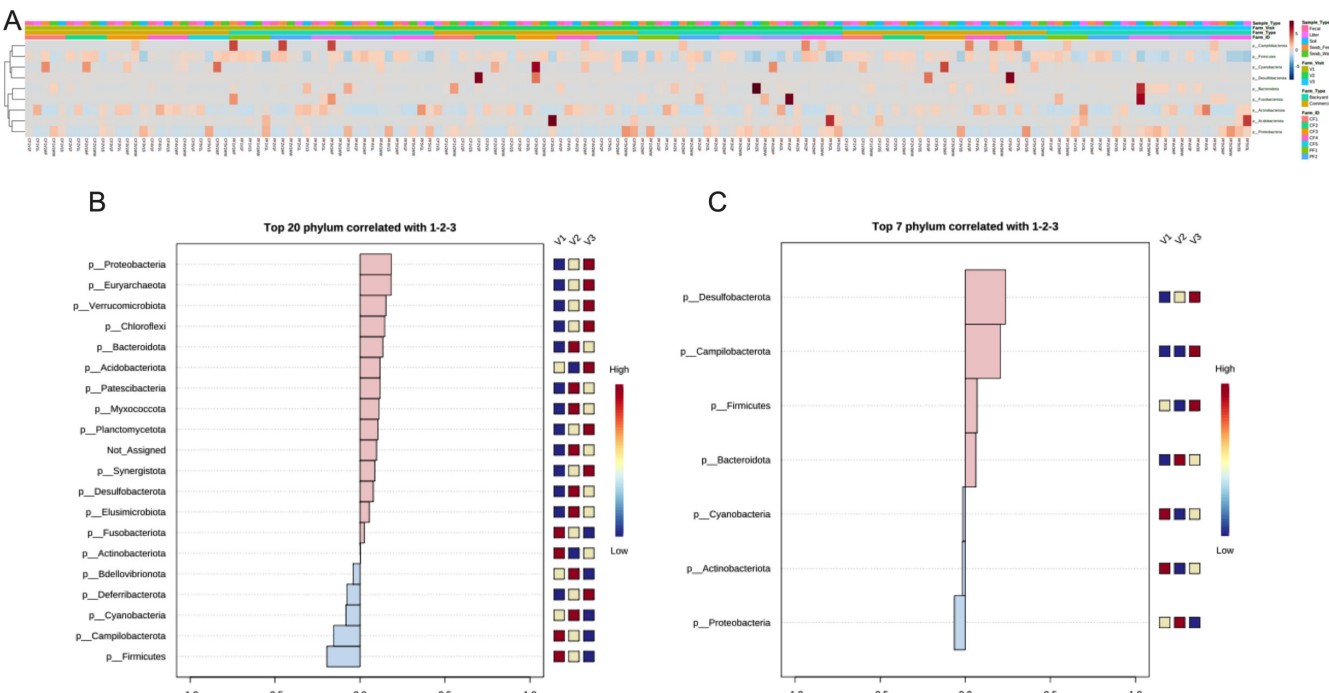

**FIG 4** Changes in microbiome profiles over time. (A) Heat map shows relative abundance of various phyla. (B) Correlation of top phyla associated with first (V1), second (V2), and third (V3) visits in backyard farms. (C) Correlation of top phyla associated with first (V1), second (V2), and third (V3) visits in backyard farms.

diversity measurements of fecal samples increased with age in both conventional and non-conventional farms (Fig. S1). The results were consistent with other studies, which reported a dramatic increase by 7 d of age (24, 25). It is well known that environmental conditions affect the fecal microbiome of chickens; however, there are very few studies looking into the effect of broiler rearing on the environment. In this study, we observed an increase in alpha diversity measurement, observed species, Chao1, and ACE higher in backyard farms compared to conventional farms. However, the Shannon and Simpson values were higher in soil samples from conventional farms than non-conventional farms. In conventional farms, huge fans used for ventilation carry dust outside, and it was shown that farm dust increases the Simpson and Shannon indexes of broiler farms (26). Numerous studies have attempted to characterize microbiome communities in poultry feces, litter, waterers, and feeders into one. However, poultry farms have distinct areas with different microbiome profiles. Feeders of commercial farms exhibited significantly higher alpha diversity compared to backyard farms. In contrast, commercial farm waterers had significantly lower alpha diversity than backyard farms. It has been noted that distinct parts of broiler farms have characteristic microbiome profiles (27). In addition, the farm's management practices such as feeding, housing, waste management, irrigation practices, antibiotic usage, use of pesticides, and rotation of livestock, also significantly impact the fecal and environmental microbiome (28). Variations in the management practices and environment of conventional and non-conventional farms may contribute to the differences in the microbiome profiles of their corresponding samples.

Depending on the broiler production system, several taxonomic groups had significantly different relative abundances. Correlation analysis reveals that members of phyla Cyanobacteria, Firmicutes, and Actinobacteria were highly correlated with commercial farms compared to backyard farms. Acidobacteriota, Campilobacterota, Bacteriodota, Fusobacteriota, and Proteobacteria were significantly higher in backyard farm settings compared to commercial farms. At the genus level, *Bacteroides* were highly abundant in the fecal samples of backyard farms compared to commercial farms. *Bacteroides* in the chicken gut are associated with increased body weight gain, low abdominal fat, improved breast muscle yield, and enhanced growth performance in chickens (29). *Bacteroides* and *Lactobacillus* were associated with high body weight gain, low abdominal fat, high breast muscle yield, and increased growth performance in chickens (29). *Desulfovibrio* was significantly higher in the backyard farms, and *Desulfovibrio* contributes to the removal of free hydrogen formed during anaerobic fermentation by consuming free hydrogen (30). *Acinetobacter* was significantly higher in backyard chicken fecal and feeder swab samples. *Acinetobacter* is pathogenic bacteria and is often associated with multidrug resistance, and multidrug resistant *A. pullorum* has been isolated from chicken meat (31, 32). Genera, *Brevibacterium* and *Brachybacterium,* and *Salinococcus* were significantly higher in the fecal, litter, feeder swabs, and waterer swabs of commercial farms. The presence of *Brevibacterium* and *Brachybacterium* was associated with low-performing broiler flocks (33). Studies conducted by Lu et al. show that *Salinococcus* was a predominant genus in broiler ceca and litter (34, 35). Furthermore, *Salinococcus* and *Atopostipes were* significantly higher in commercial farm soil, litter, and fecal samples. It is possible that ventilation could have helped the dissemination of these bacteria from inside the commercial farm to the outside environment.

Poultry microbiome research has demonstrated that the dynamic nature of microbial communities in poultry can vary significantly over time, with new species becoming more or less active and relationships between bacteria evolving. Changes to the microbiota due to stressors such as husbandry factors, feed additives, and environmental changes are also important components for understanding dynamics in the poultry microbiome (36–38). There is evidence suggesting that different production systems and dietary regimes have a pronounced effect on the composition and diversity of the microbial community within poultry (39–41). This includes shifts in major phyla present as well as particular genera associated with health outcomes (42). Recent studies have

reported various changes, such as an increase in Proteobacteria when chickens were exposed to heat stress compared to cooler conditions (43) or changes in Firmicutes abundance induced by animal genetics or rearing practices under commercial conditions (44–46). Overall, it is clear that poultry gut microbiomes undergo dynamic alterations over time that can be impacted by a variety of internal and external influences. Studying these microbe populations could help optimize the growth performance of animals while improving their welfare status. In our study, changes in microbiome composition over time were detected. Significant changes were observed, especially in backyard farms as compared to commercial farms. We observed a dynamic change in the levels of Proteobacteria and Firmicutes in backyard farms; Proteobacteria increased over time while Firmicutes levels decreased. However, Campilobacterota, which consists of the major poultry foodborne pathogen, *Campylobacter,* increased over time in commercial farm environments. Overall, the results show distinct patterns of microbial composition changes over time in commercial and backyard farms.

## Conclusion

The microbiome of poultry production systems has been increasingly recognized as a critical factor in the health and performance of birds. In this study, we aimed to explore the microbiome profiles of two different poultry production systems: conventional and outdoor-raised. We collected fecal and environmental samples from birds within each system and used high-throughput sequencing techniques to analyze the microbial communities. Our results showed significant differences in the microbiome composition between the two systems, with the conventional system having a more uniform microbiome and the outdoor-raised system having the highest diversity and abundance of potentially harmful bacteria. These findings provide insight into the impact of production systems on the poultry microbiome and have implications for optimizing bird health and performance, food safety, and public health.

### ACKNOWLEDGMENTS

This research was supported by U.S. Department of Agriculture (USDA) grant 2020–69012-31823.

### AUTHOR AFFILIATION

[1]Department of Population Health and Pathobiology, North Carolina State University, Raleigh, North Carolina, USA

### AUTHOR ORCIDs

Muhammed Shafeekh Muyyarikkandy http://orcid.org/0000-0003-3405-5440

### FUNDING

| Funder | Grant(s) | Author(s) |
| --- | --- | --- |
| U.S. Department of Agriculture (USDA) | 2020-69012-31823 | Siddhartha Thakur |

### AUTHOR CONTRIBUTIONS

Muhammed Shafeekh Muyyarikkandy, Conceptualization, Data curation, Formal analysis, Investigation, Methodology, Project administration, Software, Supervision, Validation, Visualization, Writing – original draft, Writing – review and editing | Jessica Parzygnat, Project administration, Writing – review and editing | Siddhartha Thakur, Conceptualization, Investigation, Methodology, Project administration, Supervision, Validation, Writing – review and editing, Funding acquisition, Resources

## DATA AVAILABILITY

The sequence data generated in this study were deposited in the National Center for Biotechnology Information Short Read Archive database under BioProject PRJNA988252. The SRA accession numbers and metadata are provided in the supplementary metadata file.

## ADDITIONAL FILES

The following material is available online.

### Supplemental Material

**Figure S1 (Spectrum01682-23-s0001.pdf).** Alpha diversity measurement (Chao1) of fecal samples collected from broiler chickens during multiple visits. The data set combined samples obtained from both commercial and backyard farms. The data clearly demonstrate a progressive increase in the alpha diversity measurement with each subsequent visit.

**Metadata (Spectrum01682-23-s0002.txt).** All the metadata and accession numbers of sequence data submitted as NCBI Bioproject PRJNA988252.

### Open Peer Review

**PEER REVIEW HISTORY (review-history.pdf).** An accounting of the reviewer comments and feedback.

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
