## [Reviewer comments · Microbiology Spectrum]

Microbiology Spectrum

Uncovering Changes in Microbiome Profiles Across Commercial and Backyard Poultry Farming Systems

Muhammed Shafeekh Muyyarikkandy, Jessica Parzygnat, and Siddhartha Thakur

Corresponding Author(s): Muhammed Shafeekh Muyyarikkandy, NC State University

Review Timeline:

Submission Date:	April 21, 2023
Editorial Decision:	June 23, 2023
Revision Received:	June 28, 2023
Accepted:	July 7, 2023

Editor: Blaire Steven

Reviewer(s): Disclosure of reviewer identity is with reference to reviewer comments included in decision letter(s). The following individuals involved in review of your submission have agreed to reveal their identity: Agnes Agunos (Reviewer #1); Getahun E Agga (Reviewer #2)

Transaction Report:

DOI: <https://doi.org/10.1128/spectrum.01682-23>

June 23, 2023

Dr. Muhammed Shafeekh Muyyarikkandy
NC State University
Population Health and Pathobiology
1052 William Moore Dr.
Raleigh, North Carolina 27606

Re: Spectrum01682-23 (**Uncovering Changes in Microbiome Profiles Across Commercial and Backyard Poultry Farming Systems**)

Dear Dr. Muhammed Shafeekh Muyyarikkandy:

After review of your manuscript by myself and two reviewers, the decision is that this manuscript is acceptable for publication after modification. Please see reviewer comments below.

Please also make sure that all data is accessible before resubmitting the manuscript.

Link Not Available

Sincerely,

Blaire Steven

Journals Department
Reviewer comments:

Reviewer #1 (Comments for the Author):

Introduction

Line 66: minor: antibiotics are not only used for growth promotion but for treatment, prevention and control. Also, please consider revising "antibiotic use could result in gut dysbiosis in chickens.

Materials and methods

Line 104: major - these are all required for context relevant to the results. This section requires a brief overview of the farm-level demographics, at the minimum, breed (genetics was mentioned in the discussion), the population size (number of barns, total birds raised), basic biosecurity (presence of other animals on farm) and litter/manure disposal (on-farm composting, re-used litter, hauled away?) average rearing period, cleaning and disinfection prior to chick placement (+antimicrobial use) that could potentially impact the microbiome. Was a data collection/questionnaire not used here?

As for a baseline data, was there no day 0 (chick placement) sampling (reflective of starting point microbiome status before chick placement)? Rationale for the timepoints/farm visits?

Results: 210 to 212 - these sentences belong to the discussions.

Discussions

Lines 249-250 - "so what?" aspects need to be addressed, what is the significance of this study to poultry rearing and how is this relevant to food safety and public health? Which poultry production type should be monitored more closely (rearing conditions affecting microbiome) from a food safety and public health perspective based on the predominance of certain phyla/genera?

Lines 290-291 - what are these farm management practices?

Conclusion

Line 346 to 347 - implications on food safety/public health.

Reviewer #2 (Comments for the Author):

The header "Sample collection" requires more information than what has been provided. State how the farms were selected.

Line 107-109: please clarify if the number of each sample presented here is per farm. How each sample type was obtained: more information is needed.

Line 163-164 and elsewhere: please present the statement(s) in the form of "higher" or "lower" instead, for better understanding.

Line 193-194: I think you need to include backyard and commercial farms in the statement.

Italicize all the genera names throughout the manuscript as you did 206-213.

Discussions were presented in the results and vice versa: example line 210-213.

Table 1: The asterisks (*) for significance, for some measurements it was shown in both production types; for others it was shown only for one of them. To avoid confusion, please present the "*" sign only for one of the production types choosing "higher" or "lower" and state this in the caption.

All figures have poor resolution and at times very difficult to read the texts. Please provide publication quality figures.

Figure 2: Although color keys for backyard vs commercial were presented, these were not reflected in the figure. OR please explain what the keys mean. The same for Figure 3.

Figure 4: Resolution is low. The colors for the visits not reflected in the graph. It is not clear what the bar charts indicate.

Staff Comments:

Preparing Revision Guidelines

Please return the manuscript within 60 days; if you cannot complete the modification within this time period, please contact me. If you do not wish to modify the manuscript and prefer to submit it to another journal, please notify me of your decision immediately so that the manuscript may be formally withdrawn from consideration by Microbiology Spectrum.

Response to Reviewer #1

1. Line 66: minor: antibiotics are not only used for growth promotion but for treatment, prevention and control. Also, please consider revising "antibiotic use could result in gut dysbiosis in chickens."

The sentence in the manuscript is changed accordingly.

2. Line 104: major - these are all required for context relevant to the results. This section requires a brief overview of the farm-level demographics, at the minimum, breed (genetics was mentioned in the discussion), the population size (number of barns, total birds raised), basic biosecurity (presence of other animals on farm) and litter/manure disposal (on-farm composting, re-used litter, hauled away?) average rearing period, cleaning and disinfection prior to chick placement (+antimicrobial use) that could potentially impact the microbiome. Was a data collection/questionnaire not used here?

As for a baseline data, was there no day 0 (chick placement) sampling (reflective of starting point microbiome status before chick placement)? Rationale for the timepoints/farm visits?

All these comments are addressed by adding a new section “*Farm demographics and operational practices*” under “Materials and Methods.”

Farm demographics and operational practices

The study involved sampling from 5 commercial farms and 5 backyard farms, adhering to the minimum inclusion criteria of raising at least 100 chickens without the usage of antibiotics as growth promoters. The anonymous data was safeguarded by a coding system, assuring confidentiality of farm information. Our sampling protocol varied according to the type of farm; backyard farms were visited thrice during the production cycle, specifically on days 10, 31, and 52, while commercial farms were visited on days 10, 24, and 38, reflecting the more rapid pace of broiler production in these environments. The chosen timepoints likely represent key stages in the broiler's growth and health, allowing the study to capture meaningful changes in the microbiome across their accelerated growth period. The commercial farms, employing an intensive production system, reared thousands of birds indoors, with new litter added prior to each stocking cycle. Composting was not implemented in these settings. Conversely, backyard farms which usually raise between 100-1000 birds at a time, practiced on-site composting and daily rotation of pens on the pasture. Furthermore, multiple livestock species were maintained in the backyard farms. A marked contrast was observed in biosecurity measures between the two farm types. Commercial farms implemented rigorous procedures such as tire washes, foot dips, and personal protective equipment usage. However, biosecurity practices in backyard farms were more varied, with gum boots being the most common protective gear used. Lastly, the typical production duration in commercial farms was 5-6 weeks, with backyard farms

taking a slightly longer time of 8-10 weeks. This may reflect the differences in breed genetics and production strategies between the two settings. However, data collection did not include a baseline microbiome sample at day 0 (chick placement), an aspect that could have provided valuable insight into initial conditions. The specific rationale for the chosen timepoints of farm visits pertained to the different production schedules of commercial and backyard farms.

3. *Results: 210 to 212 - these sentences belong to the discussions.*

The sentences are moved to discussion

4. *Lines 249-250 - "so what?" aspects need to be addressed, what is the significance of this study to poultry rearing and how is this relevant to food safety and public health? Which poultry production type should be monitored more closely (rearing conditions affecting microbiome) from a food safety and public health perspective based on the predominance of certain phyla/genera?*

The comments are addressed as below in the manuscript. The population of Proteobacteria in backyard farms demonstrated a rising trend over time, whereas the number of Firmicutes showed a decreasing trajectory. In commercially raised poultry, there was an upward trend in the prevalence of Campilobacterota, which includes *Campylobacter*, a primary pathogen causing foodborne diseases originating from poultry. Interestingly, a greater abundance of Bacteroides, which is associated with enhanced growth rates in chickens, was observed in backyard farms. However, a drawback was the significant increase in the concentration of potentially harmful *Acinetobacter* in samples from both backyard chicken feces and feeder swabs. Commercial farms, on the other hand, saw a marked increase in *Brevibacterium* and *Brachybacterium*, bacteria associated with broiler flocks that perform poorly. The implications of these findings can significantly impact the poultry farming industry. The increasing levels of Proteobacteria and Bacteroides in backyard farms might suggest a potential for improved growth performance in these environments. However, the higher presence of *Acinetobacter* poses a health risk that needs to be managed. On the commercial side, the increasing presence of the foodborne pathogen *Campylobacter* requires vigilance and proactive strategies to prevent potential outbreaks. The presence of *Brevibacterium* and *Brachybacterium*, indicators of low-performing broiler flocks, might suggest a need for improved farming practices to enhance the productivity and overall health of the flocks in commercial farm environments.

5. *Lines 290-291 -what are these farm management practices?*

The sentence modified to include the management practices such as feeding, housing, waste management, irrigation practices, antibiotic usage, use of pesticide, and rotation of livestock.

6. *Line 346 to 347 - implications on food safety/public health.*

Implications on food safety and public health is added to the text.

Response to Reviewer #2

1. *The header "Sample collection" requires more information than what has been provided. State how the farms were selected.*

A new section “*Farm demographics and operational practices*” under “Materials and Methods” is added.

Farm demographics and operational practices

The study involved sampling from 5 commercial farms and 5 backyard farms, adhering to the minimum inclusion criteria of raising at least 100 chickens without the usage of antibiotics as growth promoters. The anonymous data was safeguarded by a coding system, assuring confidentiality of farm information. Our sampling protocol varied according to the type of farm; backyard farms were visited thrice during the production cycle, specifically on days 10, 31, and 52, while commercial farms were visited on days 10, 24, and 38, reflecting the more rapid pace of broiler production in these environments. The chosen timepoints likely represent key stages in the broiler's growth and health, allowing the study to capture meaningful changes in the microbiome across their accelerated growth period. The commercial farms, employing an intensive production system, reared thousands of birds indoors, with new litter added prior to each stocking cycle. Composting was not implemented in these settings. Conversely, backyard farms which usually raise between 100-1000 birds at a time, practiced on-site composting and daily rotation of pens on the pasture. Furthermore, multiple livestock species were maintained in the backyard farms. A marked contrast was observed in biosecurity measures between the two farm types. Commercial farms implemented rigorous procedures such as tire washes, foot dips, and personal protective equipment usage. However, biosecurity practices in backyard farms were more varied, with gum boots being the most common protective gear used. Lastly, the typical production duration in commercial farms was 5-6 weeks, with backyard farms taking a slightly longer time of 8-10 weeks. This may reflect the differences in breed genetics and production strategies between the two settings. However, data collection did not include a baseline microbiome sample at day 0 (chick placement), an aspect that could have provided valuable insight into initial conditions. The specific rationale for the chosen timepoints of farm visits pertained to the different production schedules of commercial and backyard farms.

2. *Line 107-109: please clarify if the number of each sample presented here is per farm. How each sample type was obtained: more information is needed.*

The number of samples collected is during each visit from each farm, and the manuscript text is modified.

3. *Line 163-164 and elsewhere: please present the statement(s) in the form of "higher" or "lower" instead, for better understanding .*

The sentence in the manuscript is modified accordingly.

4. *Line 193-194: I think you need to include backyard and commercial farms in the statement.*

The sentence in the manuscript is modified accordingly.

5. *Italicize all the genera names throughout the manuscript as you did 206-213.*

All the genus names are italicized.

6. *Discussions were presented in the results and vice versa: example line 210-213.*

The sentences were moved from results to discussion.

7. *Table 1: The asterisks (*) for significance, for some measurements it was shown in both production types; for others it was shown only for one of them. To avoid confusion, please present the "*" sign only for one of the production types choosing "higher" or "lower" and state this in the caption.*

The table is modified and now * placed only on values which is significantly higher than the other production system.

8. *All Figure have poor resolution and at times very difficult to read the texts. Please provide publication quality figures.*

High resolution images are uploaded.

9. *Figure 2: Although color keys for backyard vs commercial were presented, these were not reflected in the figure. OR please explain what the keys means. The same for Figure 3.*

New figures are uploaded.

10. *Figure 4: Resolution is low. The colors for the visits not reflected in the graph. It is not clear what the bar charts indicate.*

New figures are uploaded.

July 7, 2023

Dr. Muhammed Shafeekh Muyyarikkandy
NC State University
Population Health and Pathobiology
1052 William Moore Dr.
Raleigh, North Carolina 27606

Re: Spectrum01682-23R1 (**Uncovering Changes in Microbiome Profiles Across Commercial and Backyard Poultry Farming Systems**)

Dear Dr. Muhammed Shafeekh Muyyarikkandy:

Thank you for your re-submission, I am happy to let you know that the manuscript has been accepted for publication.

Your manuscript has been accepted, and I am forwarding it to the ASM Journals Department for publication. You will be notified when your proofs are ready to be viewed.

Sincerely,

Blaire Steven
Editor, Microbiology Spectrum
